# NeuroLoom: Modeling Cortical Microcircuits with Spiking Neural Networks

**Sunkalp Chandra,**[*] **Gautham Korrapati,**[†] **Dheeraj Chintapalli,**[†] **Seungyong Yang,**[†]
**Saketh Satti**[‡] **& Jainish H. Patel**[‡]
Reteena
San Francisco, CA, United States

## Abstract

A key problem in neuroscience is understanding how cognitive function emerges from the dynamics of neural circuits. While classical Spiking Neural Networks (SNNs) provide a viable, biologically-based template for modeling these dynamics, their value can emerge only through comparisons to experimental data. Here, we present NeuroLoom, an open-source, customizable framework to design, validate, and analyze models of cortical microcircuits against experimental data. NeuroLoom provides a complete end-to-end workflow that incorporates a model library of standard neurons and synapses and programmatic access to the Allen Brain Observatory experimental recordings through the AllenSDK, making it simple to construct, fit, and analyze model microcircuits. We demonstrate the capabilities of the framework through a pilot experiment in which a cortical microcircuit model of adaptive exponential integrate-and-fire (AdEx) neurons is constrained and validated against in-vivo electrophysiology recordings from the mouse visual cortex. The framework additionally contains runnable examples of important cortical phenomena, including models of synaptic plasticity, cognitive performance, and neuromodulation, making NeuroLoom a step towards building stronger, verifiable models of the brain, and is an extensible, interpretable framework to facilitate advancing interdisciplinary research.

## 1 Introduction

A major goal in modern neuroscience is to understand how cognitive systems arise from collective neural activity (Churchland & Sejnowski, 1993; Buzsaki, 2006). While artificial intelligence produces models capable of complex tasks, these often act as "black boxes" that provide little insight into the biological algorithms underlying brain function (Richards et al., 2022). Computational neuroscience therefore aims to develop models that are biologically constrained, functionally capable, and dynamically faithful (Dayan & Abbott, 2001; Abbott, 2008). Rigorous models are validated against extensive neurophysiological datasets, including spike times, local field potentials, behavioral data, and metadata, maintaining a connection to experimental reality (Arkhipov et al., 2018).

Grounding models in data is essential to manage cortical circuit complexity, requiring a balance of excitation and inhibition (E-I balance) (van Vreeswijk & Sompolinsky, 1996; Denève & Machens, 2016), multiple cell types with intrinsic dynamics (e.g., spike-frequency adaptation) (Izhikevich, 2004), structured connectivity (Song et al., 2005), and dynamic influences from brain state, including oscillations and neuromodulation by dopamine and acetylcholine (Schultz et al., 1997; Hasselmo, 2006). A robust framework must test these features against rich, multi-modal data.

Point-neuron models such as LIF and AdEx, and plasticity rules like STDP, have long-established mathematical descriptions (Brette & Gerstner, 2005; Izhikevich, 2004). Modern challenges lie not only in model creation but in their rigorous application. There is a pressing need for frameworks that streamline validation against large-scale datasets (Crook et al., 2013), providing efficient simu-

---

[*]Corresponding author: `sunkalp@reteena.org`
[†]Equal second-author contribution
[‡]Equal third-author contribution

lation, comparable metrics, statistical analyses, and visualizations of discrepancies. Without such standardization, model development becomes fragmented, time-consuming, and labor-intensive.

## 1.1 RELATED WORK

There are computational models of neural circuits that capture a spectrum of abstraction. Rate-based models and standard ANNs have been successful in explaining high-level cognitive phenomena, but often entirely disconnect from the underpinning biophysical mechanisms, so interpretation is often poor (Wilson & Cowan, 1972; Richards et al., 2022). On the other hand, detailed multi-compartment models that capture the precise morphology of neurons can provide fantastic hierarchical detail, but the computational burden becomes the limiting factor, as they require very explicit modeling of all neurons in the network (Segev et al., 1998; Markram et al., 2015).

Spiking Neural Networks (SNNs) are in a unique space. The SNN's point-neuron models, such as the Adaptive Exponential Integrate-and-Fire (AdEx) neuron, capture the important features of neural firing and adaptation dynamics and are still feasible for large-scale simulations (Brette & Gerstner, 2005; Maass, 1997). Brian 2 is able to offer the tools to run this model; however, the big issue we continually face is that our experimental data have become rich and multi-modal (Neuropixels to name one) but few researchers are explicitly validating their SNN against their empirical data (Goodman & Brette, 2008; Steinmetz et al., 2021). We aim to bridge this gap through the development of an integrated and open-source package that combines these introductory simulation tools and a clear workflow for reproducible validation against empirical data. We are not introducing new SNN models, we are enabling you to validate the models you already have!

## 1.2 THE NEUROLOOM FRAMEWORK

We introduce **NeuroLoom**, a computational framework built on Brian 2, designed to provide a reproducible environment for constructing, simulating, and validating cortical microcircuit models against experimental recordings. NeuroLoom emphasizes both transparency and extensibility, allowing researchers to explore established neural models and assess their quantitative alignment with empirical data. The main contributions of the framework include:

- **Comprehensive SNN Validation Workflow:** NeuroLoom implements an end-to-end workflow that connects user-defined spiking neural networks (SNNs) with iterative validation using in-vivo electrophysiology datasets, such as those from the Allen Brain Observatory. By providing tools for extraction, statistical comparison, and visualization of both simulated and experimental data, the framework enables rigorous quantitative assessment of model behavior beyond qualitative inspection. This approach facilitates rapid detection of mismatches between model predictions and empirical observations and supports iterative refinement of network parameters.

- **Modular Toolkit for Cortical Modeling:** NeuroLoom provides a set of modular demonstrations that capture four core cortical phenomena:
    1. E-I balance constrained by experimental data,
    2. Cognitive task performance using simple working memory paradigms,
    3. Activity-dependent synaptic plasticity via established STDP rules, and
    4. Neuromodulatory influences (e.g., dopamine modulation of plasticity and acetylcholine modulation of excitability).

  Each module is executable and extensible, allowing users to systematically manipulate network properties, swap neuron or synapse models, and explore how interactions between different mechanisms affect emergent network dynamics. The framework thus serves as a shared resource for both hypothesis generation and validation, accelerating the cycle of model experimentation and comparison with empirical data.

## 2 METHODS

All models were implemented using the Brian 2 simulator (Goodman & Brette, 2008) and the model parameters are summarized in the Appendix.

## 2.1 Neuron and Synapse Models

NeuroLoom employs two complementary point-neuron models to balance biological realism with computational tractability. The Leaky Integrate-and-Fire (LIF) neuron primarily represents inhibitory populations and supports multi-layer network motifs for exploratory simulations (Lapicque, 1907). LIF neurons efficiently capture subthreshold integration, membrane leak, and threshold-driven spiking without detailed morphology, enabling large-scale simulations while preserving temporal fidelity for spike-timing analyses.

The primary excitatory element is the Adaptive Exponential Integrate-and-Fire (AdEx) neuron (Brette & Gerstner, 2005), which reproduces key cortical pyramidal neuron features: (i) spike-frequency adaptation via an adaptation current, and (ii) nonlinear spike initiation through an exponential term. AdEx neurons generate regular spiking, adapting bursts, and sustained-input responses, driving network dynamics through interactions with inhibitory LIF neurons to establish excitation-inhibition balance, rhythmic activity, and emergent temporal patterns.

Synaptic interactions are conductance-based, allowing postsynaptic potentials to scale with membrane potential. When plasticity is enabled, weights evolve according to standard Spike-Timing-Dependent Plasticity (STDP) (Hebb, 1949; Gerstner et al., 1996; Bi & Poo, 1998), adjusting connection strength based on spike timing. Weights are constrained within biologically plausible bounds to maintain stability during long simulations.

This dual-model approach captures diverse cortical dynamics while remaining tractable for iterative simulations, parameter exploration, and validation against experimental data. The framework supports extension with new neuron or synapse models, enabling alternative formulations or species-specific dynamics.

### 2.1.1 Multi-Layer LIF Model with STDP

To examine principles of cortical microcircuit organization, synaptic plasticity, and emergent dynamics, NeuroLoom implements a four-layer spiking network composed of leaky integrate-and-fire (LIF) neurons. The architecture is inspired by neocortical laminar structure, where intra- and inter-layer interactions support complex temporal dynamics, oscillations, and hierarchical processing (Buzsaki & Llinas, 2013). Although simplified, the model retains key mechanisms necessary for studying self-organization and the interaction between connectivity and plasticity.

Each layer contains LIF neurons partitioned into excitatory and inhibitory populations according to an anatomically motivated E/I ratio. Connectivity is dense within layers and bidirectional between layers, enabling recurrent, feedforward, and feedback interactions. This structured design provides a controlled framework for investigating how network architecture shapes activity evolution.

Synaptic weights evolve via spike-timing-dependent plasticity (STDP), modifying intra- and inter-layer connections based on precise spike timing. This rule promotes the emergence of structured activity and functional connectivity motifs consistent with cortical dynamics, including synchronized assemblies, feedforward pathways, and inhibitory stabilization (Gerstner et al., 1996; Bi & Poo, 1998). Weights are constrained within biologically plausible bounds to preserve stable firing regimes.

The model is evaluated under oscillatory and stochastic inputs to assess self-organization and activity stability. NeuroLoom supports systematic manipulation of input amplitude, frequency, and correlation structure to examine how external drive interacts with STDP to shape layer-specific dynamics. Analyses of spike rasters, firing rates, and inter-spike interval distributions quantify emergent phenomena such as phase-locking, rhythmic oscillations, and selective amplification of temporally correlated inputs.

This layered LIF network provides a tractable platform for studying how plasticity and laminar structure jointly influence cortical computation through interactions among recurrent connectivity, inter-layer communication, and activity-dependent synaptic modification.

### 2.1.2 AdEx Model for Spike-Frequency Adaptation

The NeuroLoom framework includes a module for exploring the dynamics of the Adaptive Exponential Integrate-and-Fire (AdEx) neuron in isolation, highlighting how intrinsic properties shape firing

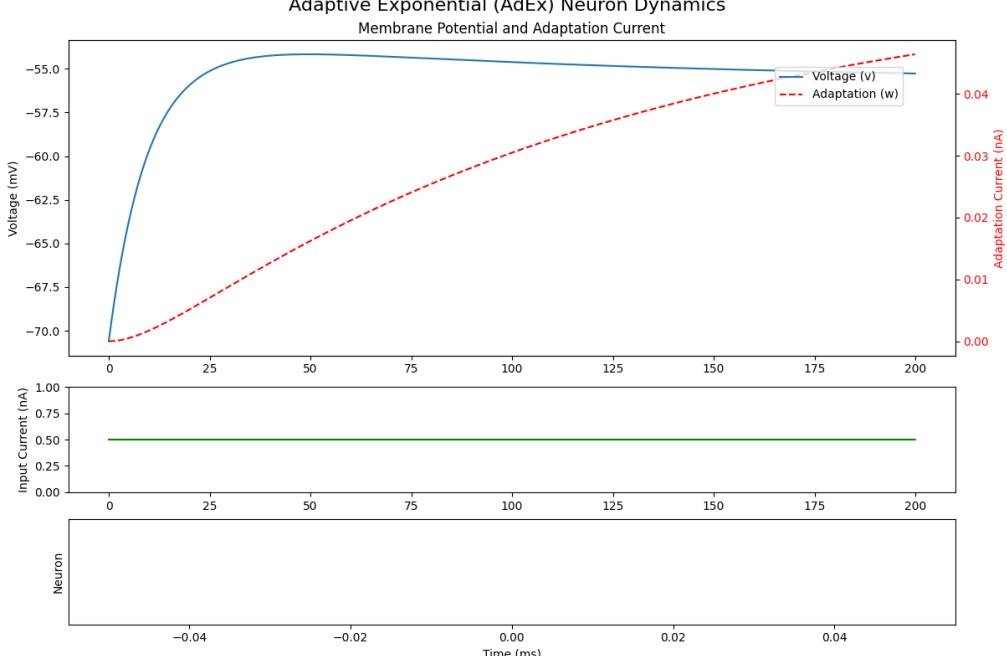

Figure 1: Demonstration of the Adaptive Exponential (AdEx) neuron model. The plot shows the membrane potential ($v$) and adaptation current ($w$) over time in response to constant input, illustrating characteristic spike-frequency adaptation.

patterns. This is particularly useful for investigating spike-frequency adaptation, where firing rate gradually declines in response to sustained depolarizing input.

In this setup, a constant current elicits rapid spikes interspersed with subthreshold oscillations. The adaptation variable ($w$), representing slow potassium conductances, accumulates during firing, exerting an outward hyperpolarizing current that reduces the instantaneous firing rate until a quasi-steady-state is reached. This reproduces key features of pyramidal neuron behavior, including initial high-frequency bursts followed by slower tonic firing.

The module also demonstrates how parameters such as adaptation time constant ($\tau_w$) and spike-triggered increment ($b$) modulate firing patterns: increasing $\tau_w$ slows recovery, enhancing spike-frequency decline, while adjusting $b$ changes the steady-state firing rate. Input amplitude, duration, and waveform can also be systematically varied to quantify input-output relationships and study interactions between intrinsic adaptation and synaptic or network-driven activity.

Although focused on single-cell dynamics, these experiments provide a foundation for understanding population-level phenomena, including rhythmic oscillations, firing-rate homeostasis, and response adaptation to sustained stimuli. Figure 1 illustrates a representative simulation, showing the characteristic decline in firing rate and adaptation current under constant input. This demonstration allows users to observe how intrinsic neuron dynamics shape temporal spike patterns, informing interpretation of more complex network behaviors.

### 2.1.3    NEUROMODULATION EFFECTS

NeuroLoom demonstrates neuromodulatory influences on neural dynamics, showing how global chemical signals shape circuit behavior without directly encoding information. Dopamine and acetylcholine exert slow, diffuse effects by modulating synaptic efficacy, intrinsic excitability, and learning rules. To illustrate these effects, the framework implements simplified, parametric models within a minimal two-neuron circuit.

In the dopamine demonstration, neuromodulation acts as a multiplicative scaling factor on synaptic plasticity updates. Dopamine regulates the magnitude of spike-timing-dependent plasticity (STDP), affecting learning rate and direction rather than instantaneous transmission. As shown in Figure 2 (top), increasing dopamine strengthens synaptic potentiation across repeated pairings, reflecting its

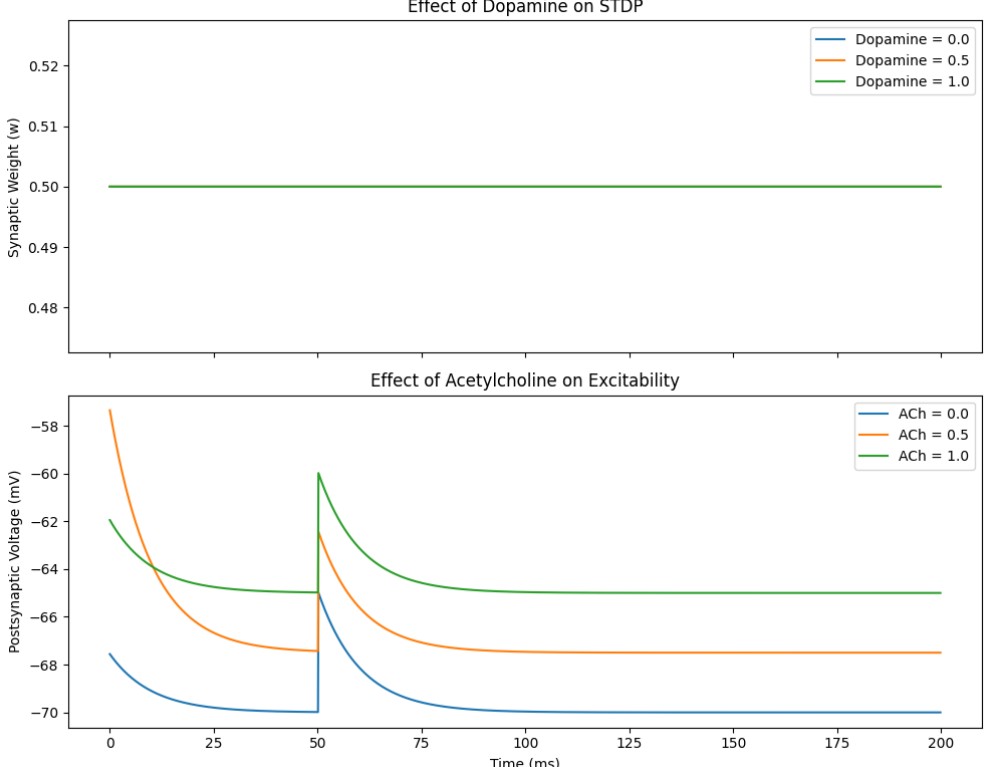

Figure 2: Simulated neuromodulation impact. Top panel: the consequence of different dopamine levels on synaptic weight over time across many episodes and during plasticity. Bottom panel: acetylcholine's effect on membrane potential excitability.

role in gating plasticity while leaving baseline transmission largely unchanged. Dopamine does not directly alter membrane dynamics or firing thresholds in this setup.

Acetylcholine, in contrast, modulates intrinsic excitability via an additive depolarizing bias to the postsynaptic membrane potential. Higher acetylcholine levels increase responsiveness to synaptic input, as seen in Figure 2 (bottom), capturing its role in enhancing cortical excitability and signal-to-noise ratio without task-specific assumptions.

These demonstrations highlight NeuroLoom's modular incorporation of neuromodulatory mechanisms. Although minimal and not fitted to empirical concentrations, they provide a framework to study how slow, global signals interact with fast synaptic and spiking dynamics.

### 2.1.4 COGNITIVE TASK PERFORMANCE

To demonstrate that NeuroLoom supports higher-level functional behavior beyond isolated biophysical mechanisms, the framework includes a 1-back working memory paradigm. This task requires the network to transiently retain a recently presented stimulus and respond based on whether the current input matches the preceding one. Despite its simplicity, the 1-back task captures a core working memory operation—short-term maintenance and comparison—and is widely used as a benchmark in experimental and computational studies (Cohen et al., 1990).

Within NeuroLoom, the task is realized through sequential stimulus-driven inputs to a spiking neural network, with performance emerging from recurrent dynamics, synaptic connectivity, and intrinsic neuronal properties rather than explicit symbolic memory states. This emphasizes cognitive function as an emergent property of distributed spiking activity.

Network activity is visualized with multiple complementary representations. A spike raster (Figure 3a) shows trial-resolved firing patterns, revealing how the network differentiates novel and repeated inputs and maintains transient stimulus-related activity.

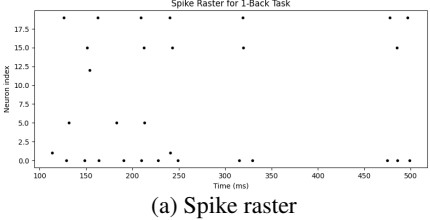
(a) Spike raster

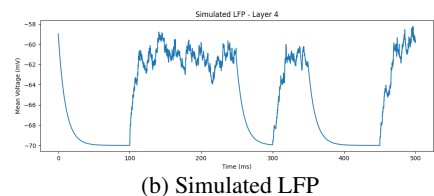
(b) Simulated LFP

Figure 3: Simulation of 1-Back working memory task. (a) Spike raster plot of neuron firing during the task. (b) Simulated Local Field Potential (LFP) showing network-level responses to task stimuli.

The framework also computes a simulated Local Field Potential (LFP; Figure 3b), providing a population-level signal to identify oscillatory components and stimulus-locked responses. The LFP bridges single-neuron activity and experimentally accessible measurements, enabling qualitative comparison with electrophysiological recordings.

Together, these results demonstrate NeuroLoom's ability to link task-level cognitive demands with biologically grounded dynamics. By integrating spike-based and population-level analyses, the framework supports investigation of how elementary cognitive functions can emerge in spiking neural networks.

### 2.1.5 COGNITIVE SIGNAL ANALYSIS

NeuroLoom includes a dedicated analysis pipeline to extract interpretable cognitive signatures from simulated neural activity. Beyond spike rasters and firing rates, it supports signal-level analyses aligned with experimental electrophysiology, enabling evaluation with standard cognitive and systems neuroscience metrics.

A central component is the simulated Local Field Potential (LFP), serving as a proxy for aggregate synaptic and population activity. NeuroLoom computes the LFP's power spectral density and quantifies power within canonical frequency bands, providing a compact representation of network dynamics across temporal scales for comparison across tasks, conditions, and model variants.

An example output is shown in Figure 4. The network exhibited a mean firing rate of 58.73 Hz, consistent with sustained, distributed activity. Spectral analysis showed uneven band power: 0.041 (Delta), 0.099 (Theta), 0.046 (Alpha), 0.056 (Beta), and 0.551 (Gamma), indicating strong fast-timescale synchronization and local recurrent interactions.

Based on these features, NeuroLoom can support heuristic inference of cognitive state. Here, elevated Gamma combined with reduced low-frequency activity was labeled a "Distracted" state. This demonstration illustrates how simulated signals can be mapped onto interpretable descriptors, and the pipeline can be extended with alternative metrics, classifiers, or experimentally grounded rules.

### 2.1.6 EMPIRICAL DATA EXPLORATION

Effective data-driven modeling requires not only experimental recordings but also tools to interrogate, summarize, and interpret them. NeuroLoom provides explicit support for exploratory analysis of empirical neural data, with a focus on large-scale datasets from the Allen Brain Observatory, bridging the gap between raw measurements and computational abstractions.

The framework includes scripts for loading, preprocessing, and visualizing multiple modalities, including spike trains, stimulus-aligned responses, and trial-averaged activity. This enables direct inspection of firing patterns, response variability, and temporal structure, allowing identification of features that should inform model design, such as baseline rates, latencies, tuning properties, and across-trial variability.

Visualizations include stimulus-aligned rasters and population-level traces, helping researchers determine whether modeled circuits should emphasize transient responses, sustained activity, oscillatory structure, or state-dependent modulation.

By integrating empirical data exploration within the framework, NeuroLoom promotes an iterative workflow grounded in observed biological statistics rather than abstract assumptions. These tools can

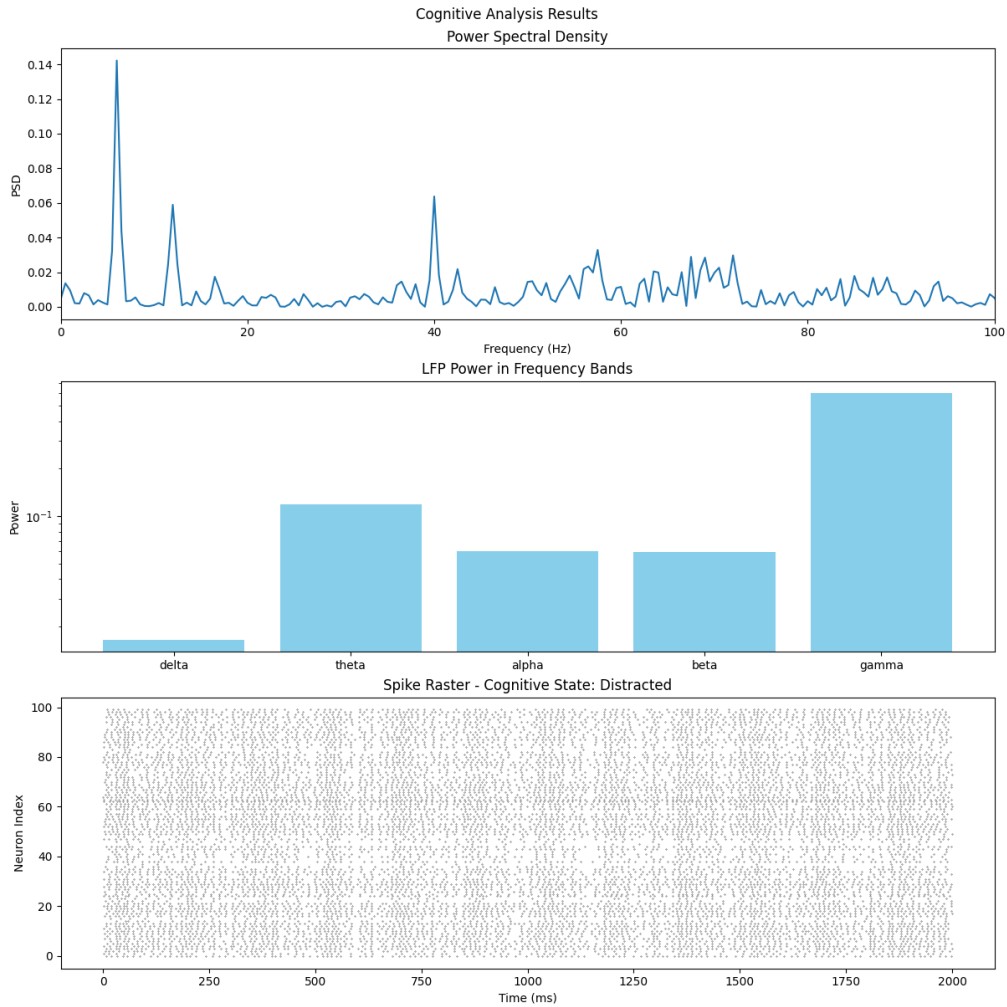

Figure 4: The cognitive signal analysis demonstration. The figure displays the (top) power spectral density (PSD) of the simulated LFP, (middle) LFP power in frequency bands, and (bottom) spike raster plot with an inferred cognitive state to illustrate the analytical capabilities of the framework.

be reused throughout model development to guide refinement as additional mechanisms or constraints are introduced.

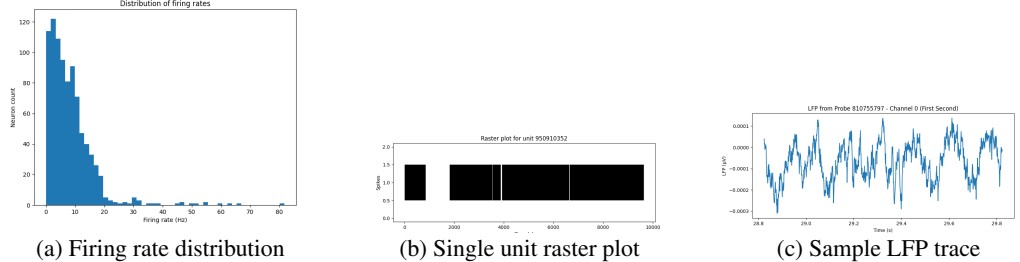

| (a) Firing rate distribution | (b) Single unit raster plot | (c) Sample LFP trace |
| --- | --- | --- |

Figure 5: Exploration of experimental data from the Allen Brain Observatory (Session ID 715093703). (a) Histogram of firing rate distribution. (b) Spike raster plot of an example single unit. (c) Sample Local Field Potential (LFP) recorded using a Neuropixels probe[cite: 132, 133, 134].

### 2.1.7 MAIN EXPERIMENT: VALIDATION AGAINST EMPIRICAL DATA

Our fundamental approach is one of acquiring data, performing a constrained simulation with scientifically validated neuron and synapse models, and conducting an in-depth comparison.

1. **Target Empirical Data & Exploration:** Using the AllenSDK, we obtained in-vivo electrophysiology data from a single Neuropixels probe in mouse visual cortex (Session ID 715093703) (Siegle et al., 2021). NeuroLoom supports raw data visualization and analysis, providing transparency in selecting validation metrics. From this session, we extracted target statistics for the model: (i) mean and standard deviation of firing rates, (ii) inter-spike interval (ISI) distributions and their coefficient of variation (CV), and (iii) power spectral density (PSD) of the local field potential (LFP), including integrated power in canonical frequency bands.

2. **Spiking Network Model:** We constructed a recurrent cortical microcircuit with 120 excitatory AdEx neurons and 30 inhibitory LIF neurons, maintaining a 4:1 excitatory-to-inhibitory ratio consistent with cortical anatomy (Meyer et al., 2010). This modest network size provides a tractable proof of concept, enabling end-to-end validation without high computational cost. External input was modeled as uncorrelated Poisson spiking with a mean rate matched to the target Allen recording. Connectivity parameters are detailed in the Appendix. The network employs standard single-neuron dynamics within a conventional recurrent architecture.

3. **Simulation and Analysis:** The model was simulated for 5 seconds, and the same statistical measures as the empirical dataset were computed. The simulated LFP was approximated as the mean synaptic current into the excitatory population, a widely used and biophysically grounded proxy (Mazzoni et al., 2008; Buzsaki et al., 2012).

## 3 EXPERIMENTS & RESULTS

NeuroLoom demonstrates neuromodulatory influences on neural dynamics, showing how global chemical signals shape circuit behavior without directly encoding information. Dopamine and acetylcholine exert slow, diffuse effects by modulating synaptic efficacy, intrinsic excitability, and learning rules. The framework implements simplified parametric models of neuromodulation within a minimal two-neuron circuit.

In the dopamine condition, neuromodulation multiplicatively scales spike-timing-dependent plasticity (STDP) weight changes, regulating learning rate and direction rather than instantaneous transmission. As shown in Figure 2 (top), higher dopamine levels progressively enhance synaptic potentiation across repeated pairings, reflecting its role in gating plasticity while leaving baseline transmission unchanged. Dopamine does not affect membrane dynamics or firing thresholds in this setup.

Acetylcholine modulates intrinsic excitability via an additive depolarizing bias to the postsynaptic membrane potential. Higher acetylcholine increases responsiveness to synaptic input, as shown in Figure 2 (bottom), capturing its role in enhancing cortical excitability and signal-to-noise ratio during attention and arousal without task-specific assumptions.

These demonstrations illustrate NeuroLoom's modular incorporation of neuromodulatory mechanisms. Although minimal and not fitted to empirical concentrations, the models provide a transparent framework to study interactions between slow global modulation and fast synaptic and spiking dynamics, allowing independent analysis and extension to larger or more detailed networks.

## 4 LIMITATIONS AND FUTURE WORK

Despite NeuroLoom's significant contributions, our primary experiment reveal limitations of the current NeuroLoom implementation:

- **Activity Mismatch:** The simulated firing rate ($0.05 \pm 0.10$ Hz) is lower than empirical observations ($8.30 \pm 8.08$ Hz), indicating a need for better calibration of E-I balance and input strengths.

- **ISI Distribution Discrepancy:** Despite qualitative similarities, the Kolmogorov-Smirnov test (statistic = 0.71, p-value = 0.0046) shows that simulated and empirical ISI distributions are statistically distinct, suggesting temporal spiking patterns need further refinement.

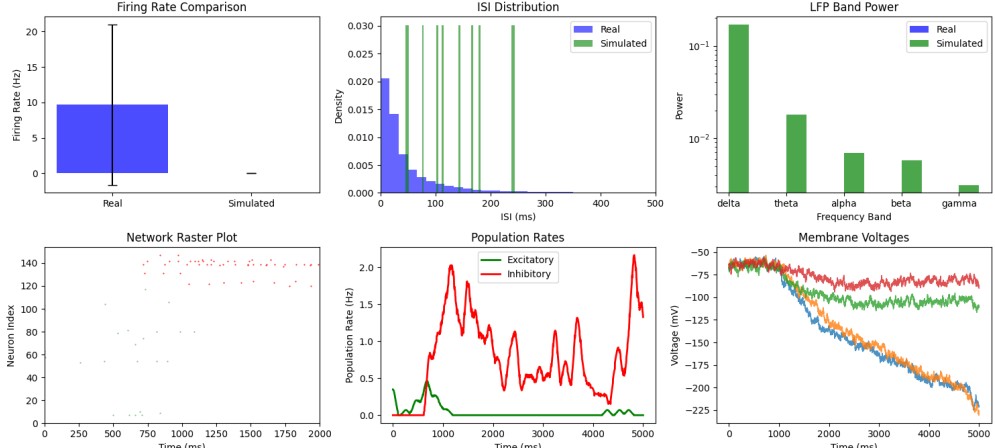

Figure 6: Comparison of main physiological measures for empirical Allen Brain Observatory data and NeuroLoom simulation. This figure provides a multi-panel comparison of (a) mean firing rates, (b) inter-spike interval (ISI) distributions, and (c) local field potential (LFP) power spectral densities across real and simulated data, showing the model's capacity to reproduce high-level statistical features

- **LFP Spectral Gaps:** The simulated LFP shows higher power in the high-gamma range and lower power in the delta range compared to empirical data, indicating an incomplete capture of slower, large-scale network dynamics.
- **Simplified Network:** The current small network (120 excitatory, 30 inhibitory neurons) and point-neuron models do not fully account for cortical complexity, including diverse cell types, detailed morphology, or extensive long-range connectivity.

To address these limitations, future work within the NeuroLoom framework will focus on:

- Performing systematic parameter exploration and optimization to better align firing rates and temporal statistics with empirical data.
- Incorporating additional, well-established neuron and synapse models while maintaining computational tractability.
- Expanding validation across multiple Allen Brain Observatory sessions and integrating richer analytical measures, including cross-frequency coupling and state-dependent spectral modulation.

## 5 CONCLUSION

We presented NeuroLoom, a reproducible framework for constructing, simulating, and validating cortical microcircuit models using spiking neural networks. NeuroLoom integrates established neuron and synapse models with programmatic access to in-vivo electrophysiology datasets, providing a complete workflow from model construction to quantitative validation. Through our experiments, we demonstrated that even small-scale networks of AdEx and LIF neurons can capture some properties of cortical circuits, including irregular spiking, spike-frequency adaptation, and network-level oscillations.

The framework's modular design allows researchers to systematically explore mechanisms such as synaptic plasticity, neuromodulation, and cognitive task performance, while maintaining reproducibility and interpretability. Our validation against empirical data highlights both the potential and current limitations of the framework: discrepancies in firing rates, ISI distributions, and low-frequency LFP power indicate areas for refinement, but also illustrate NeuroLoom's ability to identify specific targets for model improvement.

In summary, NeuroLoom unifies simulation, analysis, and validation in a single platform, offering a scalable foundation for building interpretable, empirically grounded models of cortical microcircuits. It provides a practical path forward for researchers seeking to connect mechanistic neural models with large-scale experimental data in a reproducible and systematic manner.

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

## A  APPENDIX A: MATHEMATICAL MODEL SPECIFICATIONS

### A.1  NEURON MODELS

**Adaptive Exponential Integrate-and-Fire (AdEx) - Excitatory Neurons:**

The AdEx neuron model (Brette & Gerstner, 2005) captures both spike generation and spike-frequency adaptation. The dynamics of the membrane potential $v$ and adaptation current $w$ are given by:

$$\tau_m \frac{dv}{dt} = (v_{rest} - v) + \Delta_T \exp\left(\frac{v - v_T}{\Delta_T}\right) - R_m w + R_m I_{syn} \tag{1}$$

$$\tau_w \frac{dw}{dt} = a(v - v_{rest}) - w \tag{2}$$

Reset rules: When $v \geq v_{thresh}$,

$$v \leftarrow v_{reset}; \quad w \leftarrow w + b$$

**Leaky Integrate-and-Fire (LIF) - Inhibitory Neurons:**

$$\tau_m \frac{dv}{dt} = (v_{rest} - v) + R_m I_{syn} \tag{3}$$

Reset rule: When $v \geq v_{thresh}$

$$v \leftarrow v_{reset}$$

Model Justification: AdEx is chosen for excitatory neurons because it captures spike-frequency adaptation, a key feature of cortical pyramidal neurons. LIF is selected for inhibitory neurons to reduce computational complexity while maintaining biologically plausible inhibitory control.

**Units Table:**

| Symbol | Description |
| --- | --- |
| $v, v_{rest}, v_{reset}, v_{thresh}, v_T$ | Membrane potentials (mV) |
| $w$ | Adaptation current (nA) |
| $\tau_m, \tau_w$ | Membrane/adaptation time constant (ms) |
| $R_m$ | Membrane resistance (MΩ) |
| $\Delta_T$ | Slope factor (AdEx) (mV) |
| $I_{syn}$ | Total synaptic current (nA) |
| $a, b$ | Subthreshold / spike-triggered adaptation (nS / nA) |

## A.2 Synapse and Plasticity Models

**Synaptic Current Formulation:** The total synaptic current $I_{syn}$ received by a neuron is the sum of contributions from all presynaptic neurons:

$$I_{syn} = \sum_j w_{ij} s_j(t) \tag{4}$$

where $w_{ij}$ is the synaptic weight from presynaptic neuron $j$ to postsynaptic neuron $i$, and $s_j(t)$ represents the presynaptic spike train filtered by a synaptic kernel.

**Spike-Timing-Dependent Plasticity (STDP):**

$$\tau_{pre}\frac{da_{pre}}{dt} = -a_{pre} \quad \tau_{post}\frac{da_{post}}{dt} = -a_{post} \tag{5}$$

$$a_{pre} + = A_{pre} \quad \text{on presynaptic spike} \tag{6}$$

$$a_{post} + = A_{post} \quad \text{on postsynaptic spike} \tag{7}$$

$$w + = a_{post} \quad \text{on presynaptic spike} \tag{8}$$

$$w + = a_{pre} \quad \text{on postsynaptic spike} \tag{9}$$

Plasticity Notes: When plasticity is enabled, weights are updated online according to spike timing. All synaptic weights are bounded to prevent unphysiological growth; for this implementation, we clip $w \in [0, w_{max}]$, where $w_{max}$ is specified in the parameter table.

# B Appendix B: Network Architecture and Connectivity

## B.1 Population Structure

Our cortical microcircuit model consists of two main neuronal populations: excitatory (E) and inhibitory (I) neurons. The total number of neurons in the network is:

$$N_{total} = N_{exc} + N_{inh} = 120 + 30$$

yielding an E/I ratio of 4:1, consistent with the anatomical proportion observed in mouse visual cortex (Meyer et al., 2010). This ratio ensures a balance of excitation and inhibition that preserves realistic network dynamics while keeping simulations computationally tractable.

**Population Summary:**

- Excitatory neurons (AdEx): $N_{exc} = 120$
- Inhibitory neurons (LIF): $N_{inh} = 30$

Each neuron is treated as a point neuron, and layers are abstracted as a single population (multi-layer LIF models are used in demonstration scenarios but do not affect core connectivity).

## B.2 Connectivity Rules

Connectivity between neurons follows a combination of probabilistic and distance-dependent rules. The adjacency matrix $C$ defines all connections in the network, where $C_{ij} = 1$ if neuron $i$ receives input from neuron $j$, and 0 otherwise.

- **Excitatory → Excitatory (E→E):**
  - Each excitatory neuron has a local-biased connectivity: a neighborhood of 20 neurons centered on the presynaptic neuron is defined, and 30% of neurons in this neighborhood are randomly connected.
  - Additionally, long-range sparse connections connect 5% of neurons outside the local neighborhood to each excitatory neuron.

  – This combination of local and long-range connections ensures structured excitation that mimics cortical microcircuit motifs while allowing global communication.

- **Excitatory → Inhibitory (E→I):** Each excitatory neuron connects randomly to 40% of the inhibitory population. This ensures that inhibition receives sufficient excitatory drive to regulate network activity.

- **Inhibitory → Excitatory (I→E):** Each inhibitory neuron connects randomly to 60% of the excitatory population. This enforces strong feedback inhibition to stabilize excitatory firing rates.

- **Inhibitory → Inhibitory (I→I):** Each inhibitory neuron connects to 20% of other inhibitory neurons. This allows interneuron coordination and prevents runaway inhibition.

**Synaptic Weight Initialization:** All synaptic weights $w_{ij}$ are drawn independently from a uniform distribution:

$$w_{ij} \sim U(0, w_{syn})$$

where $w_{syn} = 0.6$ mV is the base synaptic weight. Excitatory and inhibitory weights follow the same initialization but are constrained by sign, i.e., excitatory weights are positive, inhibitory weights are negative.

## C APPENDIX C: SIMULATION PROTOCOLS AND INPUTS

### C.1 NUMERICAL SIMULATION DETAILS

All simulations were performed using the Brian 2 simulator (Goodman & Brette, 2008), which provides flexible numerical integration for spiking neural networks with custom-defined neuron and synapse models.

**Integration Method:** The differential equations governing neuron dynamics (AdEx for excitatory, LIF for inhibitory) were solved using explicit Euler integration:

$$x(t + \Delta t) = x(t) + \Delta t \cdot f(x(t), t)$$

where $x(t)$ represents the state variable (membrane potential $v$ or adaptation current $w$), $f(x(t), t)$ the corresponding right-hand side of the differential equation, and $\Delta t$ the simulation timestep. Euler integration was chosen for efficiency in point-neuron models and for compatibility with Brian 2's vectorized update scheme.

**Simulation Timestep:** $\Delta t = 0.1$ ms (sufficient to resolve fast dynamics such as spike generation and adaptation currents). All time-dependent state variables updated at each timestep.

**Simulation Duration:** Each simulation ran for $T = 5$ s of biological time. This duration captures spontaneous dynamics, short-term network responses to external input, and enables robust statistics of spike trains and LFPs.

**Random Seed Policy:** To ensure reproducibility, a single fixed global random seed controlled: 1. Connectivity sampling, 2. Synaptic weight initialization, 3. External Poisson spike generation.

Table 1: Simulation parameters for numerical integration

| Parameter | Value | Description |
|---|---|---|
| $\Delta t$ | 0.1 ms | Integration timestep |
| $T$ | 5 s | Total simulation duration |
| Integrator | Explicit Euler | Numerical method |
| Seed | 42 | Global random seed for reproducibility |

### C.2 EXTERNAL DRIVE

**Poisson Input Formulation:** Each neuron receives an independent spike train modeled as a homogeneous Poisson process. Let $r_{ext}$ denote the mean rate of the Poisson input in Hz. Then the probability of a spike occurring for neuron $i$ in a timestep $\Delta t$ is:

$$P(\text{spike}_i \in [t, t + \Delta t]) = r_{ext} \cdot \Delta t$$

The inter-spike interval (ISI) distribution is exponential:

$$P(ISI = \tau) = r_{ext} \cdot e^{-r_{ext}\tau}, \quad \tau \geq 0$$

**External Input Rate Selection:** The rate $r_{ext}$ was chosen to approximately match the mean firing rate observed in the empirical Allen Brain Observatory data (8 Hz).

Table 2: External Poisson input parameters

| Parameter | Value | Description |
|---|---|---|
| $r_{ext}$ | 8 Hz | Mean input rate |
| Correlation | 0 | Inputs uncorrelated across neurons |
| Distribution | Poisson | Independent spike trains |

**Properties of the Input:** All Poisson inputs are independent and uncorrelated. Uncorrelated input ensures that network dynamics arise predominantly from internal connectivity. Input current to neuron $i$ at time $t$ is modeled as:

$$I_{ext,i}(t) = \sum_k w_{ext} \cdot \delta(t - t_{ik})$$

where $w_{ext}$ is the synaptic weight of the external input, and $t_{ik}$ are spike times from the Poisson generator.

## D    APPENDIX D: SIGNAL CONSTRUCTION AND ANALYSIS

This appendix describes the methods used to construct and analyze neural signals derived from simulations in the NeuroLoom framework.

### D.1    SPIKE-BASED METRICS

**Mean Firing Rate:** The mean firing rate of a neuron $i$ over a simulation of duration $T$ is defined as $FR_i = \frac{N_i}{T}$, where $N_i$ is the total number of spikes emitted by neuron $i$. For a population of $N$ neurons, the population mean firing rate is $FR_{pop} = \frac{1}{N} \sum_{i=1}^{N} FR_i$. Units are in Hz.

**Inter-Spike Interval (ISI) Distribution:** For each neuron $i$, the inter-spike interval (ISI) is defined as $ISI_{ik} = t_{i,k+1} - t_{ik}$. The ISI distribution is the histogram of $\{ISI_{ik}\}$, optionally normalized to represent a probability density.

**Coefficient of Variation (CV):** The CV quantifies the irregularity of spike trains: $CV_i = \frac{\sigma_{ISI_i}}{\mu_{ISI_i}}$. A CV $\approx 1$ indicates approximately Poisson-like irregular spiking.

### D.2    LOCAL FIELD POTENTIAL (LFP) APPROXIMATION

The LFP is approximated as the sum of synaptic currents impinging on the excitatory population. Let $I_{syn}^{(i)}(t)$ be the total synaptic current received by excitatory neuron $i$ at time $t$. The simulated LFP is then:

$$LFP(t) = \frac{1}{N_{exc}} \sum_{i=1}^{N_{exc}} I_{syn}^{(i)}(t)$$

This represents the aggregate postsynaptic currents, weighted equally across neurons. Units are picoamperes (pA). This approximation is widely used in computational neuroscience (Mazzoni et al., 2008).

### D.3    SPECTRAL ANALYSIS

**Power Spectral Density (PSD):** The PSD of the simulated LFP is computed using the discrete Fourier transform (DFT) of the zero-mean signal. A Hanning window is applied to reduce spectral leakage.

**Frequency Bands:**

- Delta ($\delta$): 1–4 Hz
- Theta ($\theta$): 4–8 Hz
- Alpha ($\alpha$): 8–12 Hz
- Beta ($\beta$): 12–30 Hz
- Gamma ($\gamma$): 30–80 Hz
- High-Gamma ($\gamma_H$): 80–120 Hz

**Band-Power Computation:** The power in each band $b$ is computed as $P_b = \sum_{f \in F_b} PSD(f) \cdot \Delta f$.

## E    APPENDIX E: EMPIRICAL DATA AND VALIDATION PROCEDURE

### E.1    DATASET DESCRIPTION

The empirical dataset was sourced from the Allen Brain Observatory (for Brain Science, 2020).

Table 3: Summary of experimental dataset used for validation.

| Attribute | Description |
| --- | --- |
| Session ID | 715093703 |
| Brain Region | Primary Visual Cortex (V1) |
| Species | Mus musculus (C57BL/6) |
| Recording Modality | Neuropixels extracellular probe |
| Number of Units | 112 single units |
| Sampling Rate | 30 kHz |
| Stimulus | Natural movies (30 s repeats) |

### E.2    PREPROCESSING

**Spike Sorting Assumptions:** Spike sorting was performed by the AllenSDK pipeline. Assumptions include: 1. Only high-confidence single units included. 2. Spike times aligned to 30 kHz. **LFP Preprocessing:** Raw probe voltage bandpass filtered from 1–300 Hz. Downsampled to 1 kHz. Mean subtraction applied.

### E.3    EXTRACTED TARGET STATISTICS

- **Spike-Based Metrics:** Mean Firing Rate, ISI Distribution, CV, Population-Level Firing Rate Distribution.
- **LFP-Based Metrics:** Mean LFP Trace, PSD, Band-Limited Power.

**Justification:** These metrics capture essential features of both single-neuron dynamics and population-level oscillations.

## F    APPENDIX F: COGNITIVE AND NEUROMODULATORY DEMONSTRATIONS

Code for all demonstrations is available at `https://github.com/Reteena/CortiNet`.

### F.1    COGNITIVE TASK: 1-BACK WORKING MEMORY

**Task Structure:** 1. Stimulus sequence: discrete items presented in 500 ms windows. 2. Task requirement: identify if current stimulus matches immediately preceding stimulus. 3. Input encoding: Poisson spike train projected to excitatory subset. **Neural Readout Assumption:** Summing spikes from a small population of readout neurons. **Purpose:** Illustrates how point-neuron SNNs can encode and maintain a short-term memory trace.

## F.2 NEUROMODULATION MODELS

**Dopamine (Plasticity Scaling):** Dopamine modulates the learning rate of STDP: $\Delta w = \eta \cdot D \cdot a_{pre/post}$. Increased dopamine increases potentiation magnitude. **Acetylcholine (Excitability Modulation):** Acetylcholine modulates postsynaptic membrane potential: $v(t) \leftarrow v(t) + \alpha \cdot A$. Effect: neurons fire more readily without changing underlying synaptic weights.

## G APPENDIX G: PARAMETERS, STATISTICS, AND REPRODUCIBILITY

### G.1 PARAMETER TABLE

Table 4: Condensed key simulation parameters.

| Parameter | Description | Value |
|---|---|---|
| $N_{exc}$ | Excitatory neurons | 120 |
| $N_{inh}$ | Inhibitory neurons | 30 |
| $\tau_m$ | Membrane time constant | 25/15 ms (exc/inh) |
| $v_{rest}$ | Resting potential | -65 mV |
| $v_{thresh}$ | Spike threshold | -50 mV |
| $\tau_w$ | Adaptation time const | $80 \pm 40$ ms |
| $w_{syn}$ | Synaptic weight | 0.6 mV |
| $dt$ | Integration timestep | 0.1 ms |

### G.2 REPRODUCIBILITY STATEMENT

All code, parameters, and example scripts are available at the repository. Deterministic behavior is ensured via fixed random seeds (np.random.seed(42), brian2.seed(42)). Scripts automatically execute preprocessing, simulation, and metric extraction.

