# OpenReview forum: "NeuroLoom: Modeling Cortical Microcircuits with Spiking Neural Networks"
_ICLR.cc/2026/Workshop/FM4Science — ICLR 2026 Workshop FM4Science Poster_

### Official Review · Reviewer_4ABT · 2026-02-15
**This work proposed an open-source, customizable framework for the design, validation, and analysis of cortical microcircuit models using Spiking Neural Networks (SNNs). Built upon the Brian 2 simulator, the framework’s primary contribution is a standardized end-to-end workflow that integrates programmatic access to the Allen Brain Observatory via the AllenSDK. The authors demonstrate the utility of NeuroLoom by constructing a microcircuit of Adaptive Exponential Integrate-and-Fire (AdEx) and Leaky Integrate-and-Fire (LIF) neurons, which they then validate against in-vivo electrophysiology from the mouse visual cortex. The framework includes modular demonstrations of excitatory-inhibitory (E-I) balance, synaptic plasticity (STDP), and neuromodulation, offering a "verifiable" approach to understanding how cognitive functions emerge from neural dynamics.**

**Rating:** 7
**Confidence:** 4

**Review:**

Overall the work is significant and the work provides a bridge between high-level computational abstractions and empirical biological data. The authors provide a clear breakdown of the neuron models (AdEx for excitatory and LIF for inhibitory populations) and justify these choices based on computational tractability and biological fidelity.

Strengths:
* AllenSDK Integration: Provides seamless access to standardized, large-scale neurophysiology data, making it easier for theorists to ground their models in reality.
* AdEx Model Fidelity: The choice of AdEx neurons allows for the simulation of characteristic spike-frequency adaptation, initial high-frequency bursts, and steady-state tonic firing.
* Modular Toolkit: Includes pre-built modules for neuromodulation (dopamine/acetylcholine) and plasticity (STDP), allowing users to swap components easily.
* Standardized Metrics: Computes comparable metrics like the Coefficient of Variation (CV) and Power Spectral Density (PSD) of the LFP, facilitating rigorous statistical analysis.

Weakness:
* Pilot Experiment Performance: The model currently fails to accurately match empirical firing rates and LFP spectral power in the delta range, indicating that the baseline parameters require significant manual or algorithmic optimization.
* Small Network Size: The pilot uses only limited neurons; scaling this to tens of thousands of neurons while maintaining programmatic simplicity remains an open challenge.
* Simplified Connectivity: While it uses probabilistic and distance-dependent rules, it lacks the detailed laminar specificity (layers 2/3, 4, 5, 6) found in more complex cortical models like the Potjans-Diesmann circuit

---

### Official Review · Reviewer_ibUB · 2026-02-22
**NeuroLoom is a useful idea, but the validation fails and the paper overclaims**

**Rating:** 3
**Confidence:** 4

**Review:**

NeuroLoom packages a Brian2 + AllenSDK workflow for building cortical microcircuit SNNs and comparing them to Allen recordings. The core validation experiment fails on first-order statistics, yet the paper concludes the model captures key empirical properties, so I do not think it meets the bar as written.

The paper presents NeuroLoom, an open-source framework for constructing spiking microcircuits and validating them against Allen Brain Observatory data (example: session 715093703, mouse V1, Neuropixels). It includes demos (plasticity, neuromodulation, 1-back, and a “cognitive signal analysis” pipeline).

Strengths?

Clear motivation. standardise “simulate + compute metrics + compare to data” workflows.

Useful engineering choices for a tutorial-style framework (fixed seed, explicit metrics definitions).

Practical integration point, AllenSDK-based loading and exploratory plots/scripts are valuable to many users.


Weaknesses?

The main validation result is a failure. Simulated firing rate is 0.05 ± 0.10 Hz vs empirical 8.30 ± 8.08 Hz; ISI distributions are significantly different (KS statistic 0.71, p-value 0.0046). These are not small gaps.

The conclusion overclaims. It states small AdEx/LIF networks “capture key statistical properties”, while the paper’s own results show large mismatches.

Reproducibility is used as a selling point, but the evidence is thin. Validation is one session, one fixed seed, and a 5 s simulation. No variance across seeds, no robustness across sessions.

Setup inconsistency is not diagnosed. External input rate is set to match empirical mean (~8 Hz), yet the network outputs near-silence. This strongly suggests incorrect regime (E/I balance, input weights, thresholds, connectivity), but the paper does not analyse why.

Signal proxy story is inconsistent and under-cited. Appendix defines LFP as mean synaptic current into E cells and still contains a missing citation “(?)”. The dataset description also has “(?)”. Clean citations matter in a framework paper.

“Distracted” cognitive-state labeling is not acceptable as presented. It is explicitly heuristic and unvalidated, yet framed as an interpretable descriptor. This risks misleading readers.

Double-blind issue. The PDF says double-blind, but the appendix provides a GitHub repository link and a reproducibility statement referencing “the repository,” which likely deanonymises authors.


What I would require for acceptance?

A working validation example. Show at least one parameter setting where firing-rate distribution and basic ISI statistics are reasonably aligned with the target session (not perfect, but same order of magnitude and no obvious collapse).

Minimal robustness. Repeat validation across multiple Allen sessions (at least 3–5) and across multiple seeds, reporting mean and variance.

Parameter calibration story. Even a coarse search or simple optimisation loop is fine, but “set rext = 8 Hz” and then failing is not enough.

Fix the signal proxy and citations. Remove “(?)” placeholders, and clearly state limitations of the LFP proxy relative to extracellular voltage preprocessing.

Remove or relabel the “Distracted” demo. If kept, label it as a toy example and avoid any cognitive interpretation claims.

Fix anonymisation. Use an anonymised repository or remove identifying links for review.


Questions for the authors?

Why does the network produce ~0 Hz output given an external drive chosen to match empirical mean rate? Which parameter(s) dominate this collapse?

Do you implement any fitting/optimisation beyond selecting rext? If yes, describe it precisely.

Can you report results across multiple sessions and seeds, or is the current pipeline only demonstrated for a single run?

---

### Decision · Program_Chairs · 2026-03-03

Accept (Poster)